# Beyond Behaviour: How Health Inequality Theory Can Enhance Our Understanding of the ‘Alcohol-Harm Paradox’

**DOI:** 10.3390/ijerph18116025

**Published:** 2021-06-03

**Authors:** Jennifer Boyd, Clare Bambra, Robin C. Purshouse, John Holmes

**Affiliations:** 1School of Health and Related Research, The University of Sheffield, S1 4DA Sheffield, UK; john.holmes@sheffield.ac.uk; 2Population Heath Sciences Institute, Faculty of Medical Sciences, Newcastle University, NE2 4HH Newcastle upon Tyne, UK; clare.bambra@newcastle.ac.uk; 3Department of Automatic Control and Systems Engineering, The University of Sheffield, S1 3JD Sheffield, UK; r.purshouse@sheffield.ac.uk

**Keywords:** alcohol, alcohol-related harm, socioeconomic position, health inequality, social determinants

## Abstract

There are large socioeconomic inequalities in alcohol-related harm. The alcohol harm paradox (AHP) is the consistent finding that lower socioeconomic groups consume the same or less as higher socioeconomic groups yet experience greater rates of harm. To date, alcohol researchers have predominantly taken an individualised behavioural approach to understand the AHP. This paper calls for a new approach which draws on theories of health inequality, specifically the social determinants of health, fundamental cause theory, political economy of health and eco-social models. These theories consist of several interwoven causal mechanisms, including genetic inheritance, the role of social networks, the unequal availability of wealth and other resources, the psychosocial experience of lower socioeconomic position, and the accumulation of these experiences over time. To date, research exploring the causes of the AHP has often lacked clear theoretical underpinning. Drawing on these theoretical approaches in alcohol research would not only address this gap but would also result in a structured effort to identify the causes of the AHP. Given the present lack of clear evidence in favour of any specific theory, it is difficult to conclude whether one theory should take primacy in future research efforts. However, drawing on any of these theories would shift how we think about the causes of the paradox, from health behaviour in isolation to the wider context of complex interacting mechanisms between individuals and their environment. Meanwhile, computer simulations have the potential to test the competing theoretical perspectives, both in the abstract and empirically via synthesis of the disparate existing evidence base. Overall, making greater use of existing theoretical frameworks in alcohol epidemiology would offer novel insights into the AHP and generate knowledge of how to intervene to mitigate inequalities in alcohol-related harm.

## 1. Introduction

Systematic socioeconomic inequalities in health persist and continue to widen across the globe, including in countries ranked highly on indices of economic prosperity and human development [1,2]. Alcohol-related health outcomes are not only an example of health inequality but also contribute to inequalities in both life expectancy and death age between socioeconomic groups [3]. There is a large body of evidence to suggest that, although those of lower socioeconomic position (SEP) tend to drink the same or less on average as those in higher SEPs, they still experience greater rates of alcohol-related harm [4]. One record linkage study found that, despite controlling for alcohol consumption and other risk behaviours, the most deprived group still maintain a three-fold higher risk of alcohol-related harm [5]. This phenomenon, termed the alcohol-harm paradox (AHP), treats alcohol use as a risk factor for health-related harm, although when alcohol use crosses into alcohol dependence the social/health state of dependence that arises is viewed as a harm outcome [6]. The AHP is found consistently across several outcomes, including alcohol dependence [7], alcohol-related morbidity [8] and mortality [4]. Yet the causal mechanisms remain unclear.

Despite socioeconomic inequalities in alcohol-related health outcomes, health behaviour has been central to research investigating the AHP [9]. This reflects wider public health trends, as for decades epidemiological research has been criticised for its emphasis on using individual-level proximal risk factors to predict population-level health [10,11]. Arguably this has led to the most affluent reaping the health benefits due to their increased access and uptake of behaviour change interventions [2].

Cross-sectional research has demonstrated that low SEP groups tend to drink on fewer occasions but drink more heavily per occasion compared to high SEP groups [12,13]. They are also more likely to engage in multiple health-risk behaviours (e.g., smoking, poor diet) [12]. However, these studies do not measure harm outcomes.

Conversely, two record-linkage studies found that behavioural factors, including drinking pattern, smoker status and BMI, could not fully explain the paradox [5,14]. These factors attenuated inequalities, but low SEP groups still had a persistently higher risk of alcohol-related harm [ibid.]. These findings were confirmed by a recent meta-analysis, which found that quantity of alcohol consumed and drinking patterns could not explain socioeconomic inequalities in the relative risk of both all-cause and alcohol-attributable mortality [4]. This suggests health behaviours are unable to fully explain the AHP.

While empirical research on the AHP has been limited in exploring other factors associated with socioeconomic circumstances [9], there is an increasing appetite to draw on explanations used to understand health inequalities. A report summarising the AHP discusses access to healthcare and material resources as potential explanations [15]. However, at present, there is a lack of theoretical structure to research investigating the AHP. Our understanding of the causes of the paradox remains stagnant due to a continual focus on individual behaviour. This is reflected in recent calls for exploration of contextual factors (e.g., characteristics of drinking environment) and how they not only influence health behaviour but may also directly impact harm [16].

The aim of this paper is to address this gap by identifying alternative approaches rooted in health inequality theory which could be used to design future research on the AHP. To achieve this, we review theories of health inequality and their potential to understand the causes of—and therefore potential solutions to—the AHP. We do not aim to synthesise these theories or recommend any one theory. In the context of the AHP, drawing on any of these approaches would be a novel way to conceptualise the problem or inform research design. In Section 2 we introduce prominent theories including the social determinants of health (SDH), fundamental cause theory (FCT), the political economy approach and the eco-social model and discuss the extent to which these approaches are present in the existing AHP literature. We do so by explicitly drawing on a recent systematic review which presents an overview of the explanations for the AHP [9]. We then examine how these theories could be used to explicitly frame research on the AHP. In Section 3 we discuss the potential use of computer simulations to assess their explanatory value. In Section 4 we discuss what adopting a health inequality lens could mean for the wider alcohol-harm research agenda.

## 2. Drawing on Theories of Health Inequality to Understand the AHP

Since the publication of the UK Black Report on Inequalities in Health [17], several theories have been developed which seek to explain how SEP drives health outcomes. Most have a common focus: to shift attention away from the individual-level and behavioural factors, and instead take a multi-level approach. In this section we outline four main theoretical approaches: the SDH, FCT, the political economy approach and the eco-social model (see Table 1 for descriptions of each theory), and referring to a recent review [9], discuss how these approaches fit with explanations for the AHP used within scientific literature. The review highlighted which explanations had remained hypothetical, and which were present in the empirical research. We aim to highlight how explicitly drawing on theories of health inequality could support research aiming to identify the causal mechanisms that drive the AHP.

### 2.1. The Social Determinants of Health: Current Evidence and Future Directions

The SDH refers to the social and economic factors that shape health at the individual and population level [28,29]. This approach, originated from the Rainbow Model [30], was refined by the WHO in the 2000s [18] and continues to be central to public health research. It attempts to shift the focus from individual-level behaviours as the cause of health inequality to social determinants which themselves determine not only health but also behaviour [31]. Drawing on this theoretical approach to understand the AHP could be the first step to shift from an individual approach.

Within the SDH theoretical approach there are four underlying-interrelated-explanations: culture-behaviour, materialist, psychosocial and lifecourse [32].

***Culture-Behaviour***. Norms and cultural practices associated with socioeconomic groups have been hypothesised to impact alcohol-related harm. There has been discussion, but no formal hypothesis testing, for how normative differences in drinking patterns between socioeconomic groups might contribute to AHP [33,34]. Culture and norms may also influence help-seeking and engagement with preventative healthcare services [35]. There is further scope to examine occasion-level risk factors, such as drinking contexts and their association with acute alcohol harm [36].

**Materialist**. The materialist approach is not present in empirical work investigating the AHP. Researchers have hypothesised that some of the mechanisms associated with materialism lead to socioeconomic inequalities in alcohol-related harm without explicitly drawing on theory. This includes individual material deprivation (e.g., housing and employment), which results in individuals having worse health and a lack of resources to protect themselves from a problem or stressful life event [15,37,38]. Additionally, place-based materialist mechanisms, such as a lack of environmental resources (e.g., treatment facilities and preventative services), alcohol outlet density and barriers to accessing healthcare have all been hypothesised contribute to the AHP [39,40].

When providing materialist explanations for the AHP, researchers tend to focus on the mechanisms which impact the most deprived in society, without considering the material advantages available to wealthier socioeconomic groups. Additionally, material explanations are typically discussed in isolation in the AHP literature, meaning the link between materialist explanations and societal structures (e.g., the welfare state and benefits system) is missing from the current narrative.

Historically, the contribution of individual material factors (car ownership) to health inequalities has been well evidenced [41,42,43]. Subsequent research also includes resources available in the environment (e.g., access to destinations, transportation systems) [44]. Applying these measures to identify material differences within and between socioeconomic groups could reveal the contribution of material mechanisms to the AHP.

***Psychosocial.*** The psychosocial approach has yet to be used in research investigating the AHP. Stress-related mechanisms are hypothesised to play a role, particularly lower socioeconomic groups experience a greater number of stressful life events, negative stereotyping, stigma, and social isolation [7,37,45,46]. The lack of social relationships is purported to lead to maladaptive coping strategies, consuming alcohol to cope and a reduced resilience to future negative events [47,48]. Conversely, it is acknowledged that affluent individuals have a beneficial network of social connections and therefore a greater social ‘buffer’ against stressful life events [40,49]. While these hypothesised mechanisms touch on components of the psychosocial approach, the role of social comparison (when lower socioeconomic groups compare themselves with others) and discussion of the biological consequences, both central to the psychosocial approach, are missing from the current AHP literature.

Explicitly using the psychosocial approach would reframe the discussion of psychological and social mechanisms to consider how people feel compared to others and the psychological and biological consequences of those feelings which may contribute to the inequalities expressed within the AHP. This concept of relative deprivation is particularly important given the presence of the AHP in high income social welfare state countries [9] where social inequality persists. There is a vast literature on psychosocial pathways, which have been shown to contribute to health inequalities more generally [50], particularly in the form of social capital (capturing both social buffer and potential negative effects of social inequality and exclusion). Future work aiming to understand the AHP could usefully refer to the measures of social capital used in existing studies.

***Lifecourse***. The life-course explanation integrates aspects of several other explanations, allowing different causal mechanisms and processes to explain socioeconomic health inequalities. Risk factors associated with other SDH explanations have been situated in time by some researchers investigating the AHP. This work shows promise, with one study finding cumulative behaviours (those that persist over time) attenuate the link between SEP and all-cause mortality by 38–77% compared to adjusting for proximal behaviours which attenuated the link by only 24–55% [51]. Some literature on the AHP discusses the impact of experiencing material disadvantage at critical time periods (e.g., childhood) and the accumulation of negative events as having prolonged negative health effects [52,53,54].

The life-course perspective has been adopted in research using event history analysis and retrospective data, for example in a study investigating the role of cultural capital and cultural health capital during childhood in the uptake of mammography in later life [55]. There is a lack of application of these methods in the context of alcohol-harm, with only one similar example identified in the review which investigated factors associated with the development of a comorbid alcohol and mental health condition [53].

The overall SDH approach is, however, subject to criticism. It has been argued that those who adopt it remain focused on the intermediary causes of health inequalities despite the consensus that it is the macro-level structures that result in health inequality [56]. These macro-level structures are viewed as being outside individual control and have become ‘causes of causes’ obscured by more proximate factors (e.g., health behaviour). This has resulted in theoretical and empirical research dedicated to describing the mechanisms that link socioeconomic inequalities to health, as opposed to identifying the source of socioeconomic inequality [56]. One theory developed to address this gap is FCT [22,57].

### 2.2. Fundamental Cause Theory: Current Evidence and Future Directions

FCT shifts the focus from individual-level causes of health inequalities to looking at the context; what puts people “at risk of risks” [22]. This means acknowledging that risk factors (e.g., alcohol consumption) are generated by social conditions, specifically the socioeconomic stratification of society. Crucially this theory does not deny the role of social determinants but suggests that base mechanisms associated with SEP determine whether individuals can adapt to the introduction of new disease, risks or treatment [58]. Proponents of this theory highlight that SEP should be viewed as the fundamental cause of health inequality and any downstream risk factors rooted within it [58]. From this perspective, neglecting the social conditions which generate risk factors has slowed progress in reducing health inequalities.

FCT is not apparent in research investigating the AHP. Using this theory to frame the mechanisms underlying the paradox requires a focus on the societal structures which generate social inequality. Viewing SEP as a fundamental cause of health inequality requires the understanding that disparities are generated through multiple intervening risk-factor mechanisms which alter over time [22]. Key to this is the role of resources (money, knowledge, power, prestige, and access to social connections), closely linked to the materialist approach [22,58]. FCT asserts that health inequalities will remain despite societal and healthcare changes so long as the socioeconomic structure giving access to resources remains stable [58]. Drawing on this perspective to understand the AHP would require acknowledging the existence of this structure and treating SEP as a meta-mechanism responsible for access to resources which could mitigate the effects of other factors associated with the SDH.

A comparative case-study using FCT predicted that as lung cancer becomes more preventable, due to knowledge of the link between smoking and the disease, those with greater access to resources disproportionately benefit, thus increasing health inequalities [59]. Contrastingly, for a disease lacking in major prevention or treatment innovation (e.g., pancreatic cancer), there was found to be no mortality advantage associated with socioeconomic group, and this trend was consistent across time [ibid.]. Alcohol-related harms (e.g., liver disease), are largely preventable. Trend analysis could test the role of FCT and investigate whether the introduction of prevention or treatment measures over time has resulted in socioeconomic inequalities in alcohol-related harm.

### 2.3. The Political Economy of Health: Current Evidence and Future Directions

Sitting between the SDH and FCT is the political economy of health approach. The political economy explanation is an attempt to acknowledge the role of upstream factors in generating and distributing risk factors. It argues that the social- and behavioural- determinants of health are themselves shaped by structural determinants: politics, the economy, the (welfare) state, political institutions, the organisation of work and the structure of the labour market [60,61,62] and that population health is shaped by the “social, political and economic structures and relations” that may be, and often are, outside the control of the individuals they affect [25,27].

Structural influences within the political economy approach have only been tenuously linked to the AHP. The economic and socio-political conditions, alcohol policy, corporate influence, employment, and power relations are provided as potential explanations for the AHP [45,63,64,65]; however, authors do not clearly articulate the underlying mechanisms. They touch on the commercial determinants of health as key drivers of alcohol-related harm which aligns with recent calls to acknowledge the detrimental role of the private sector on both the environment and health behaviour, which in turn determines health [66]. The political economy perspective clearly defines the role of these structures as influencing the distribution of the other SDH. Drawing on a synthesis of these perspectives in the context of alcohol-related harm would highlight these mechanisms. For example, the social and political attitudes of residents and decision makers influence the investment of public services in deprived areas, which then determines the availability of services [67], a materialist determinant of health.

Studies investigating the role of political economy in the generation of health inequalities typically take a cross-national comparative approach. This involves comparing different economic and political systems to understand how these systems contribute to health inequalities, both within and between countries [68]. This approach to research provides the opportunity to identify how the structure of the labour market, employment and welfare systems can prevent or increase health inequalities [ibid.]. There is a current lack of cross-national comparisons in the existing AHP literature.

### 2.4. The Eco-Social Model: Current Evidence and Future Directions

A recent commentary by Bloomfield has called for future research investigating the AHP to draw on the eco-social approach, acknowledging that inequalities in alcohol-related harms cannot be explained by drinking patterns alone [16]. The main distinguishing feature of the eco-social approach is the emphasis it places on biological and ecological analysis [69].

Biological mechanisms have been hypothesised to contribute to the AHP. Primarily these have been related to health behaviours and genetic alterations due to the experience of disadvantage [12,33]. For example, engaging in certain patterns of behaviour (e.g., multiple unhealthy behaviours or drinking with meals) has metabolic effects which compound or protect against the effects of alcohol consumption [33]. Biological alteration related to the experience of disadvantage or differences based on ethnicity were also more vaguely linked to the AHP [48].

Explicitly using the eco-social approach would shift the focus to how individuals biologically embody their social conditions. Achieving this in empirical research requires access to biological and social data. A recent paper which analysed data from several cohort studies investigated the relationship between social disparity and biology, finding evidence of biological changes in response to the environment [70]. There may be opportunities for alcohol researchers to engage in collaborative projects or gain access to data sets, for example, the UK Biobank [71], which would allow the opportunity to investigate the eco-social model in the context of alcohol harm.

## 3. Computer Simulations Can Test the Explanatory Value of Mechanisms Specified in Health Inequality Theory

Explicitly drawing on existing theories of health inequality may address the gap in identifying and extracting relevant variables and relationships in the pursuit to understand the AHP. However, the methods best placed to test these causal relationships requires further scrutiny.

To study these complex relationships, which exist on a multi-level plane (e.g., individual, community, and structural levels) and are dynamic in nature, suitable research methods are required. The “risk factor” approach to epidemiology explores decontextualized and independent relationships between dependent and independent variables and uses linear reductionist models to test these relationships [72]. To capture the features of complexity, a mechanism-based approach is required which explicates the details of how regularities are brought about rather than focusing on statistical regularities between variables [73]. Mechanisms consist of “entities” and the “activities” entities engage in, either as a collective or independently, to bring about a particular outcome [ibid.]. Computer simulation methods are a good candidate to test mechanisms, and complex system models have become increasingly attractive in public health research [74].

A review of the use of simulation models in the context of health inequality concluded that they enhance our understanding of socioeconomic health inequalities [75]. Specifically, the class of techniques known as agent-based modelling (ABM) can flexibly model the multilevel, reciprocal, and indirect effects of socioeconomic inequalities [ibid.]. ABMs are computer simulations comprised of agents (e.g., individuals or households) and their interactions within the context of their environment [76]. ABMs provide the opportunity to test mechanisms specified in theory [77]. This ranges from abstract theory testing to more concrete applications which draw on empirical data to inform the properties and environments of agents [78]. Much like other types of simulation model, ABMs enable otherwise fragmented evidence to be synthesised to address research questions and inform decision making [79].

One example of an ABM implemented to understand socioeconomic health inequalities explored the role of bounded rational choice mechanisms (individual level) and spatial segregation (structural level) in the emergence of income gradients in healthy eating [80]. This model represented both food stores and households as having agency over decisions to supply and purchase, respectively, healthy, or unhealthy foods. The model equations define the mutual interactions between stores and households, enabling feedback loops to be represented. The model’s findings suggest that differences in diet between socioeconomic groups arises only when high income households and healthy stores are both spatially segregated from low-income households and unhealthy stores. Once established, these diet inequalities could only be overcome when both groups had favourable preferences for healthy foods and when healthy food was relatively cheap [ibid].

A similar approach could be taken to investigate the mechanisms specified in health inequality theory. Here, we sketch such a model. In psychosocial theory, one mechanism proposed to result in socioeconomic differentials in health is that high SEP groups have a protective social buffer [40,49]. Hypothetically, this mechanism could be represented in an ABM by simulating individuals as agents and defining a macro-level social network structure with connections based on agent attribute similarity (e.g., age, gender). Agents would possess the capability to give or receive support in the presence of a stressful event. However, this capability would be contingent on their own resources (e.g., income), type of support available to them (e.g., emotional support) and their own stress burden. Individuals who receive support from their network would have a reduced stress burden and therefore reduced risk of harm. The network could also be responsive to changes in relationships (e.g., providing support strengthens ties while refusing support breaks social ties between agents). A simulation such as this would allow in silico experimentation with changes in resources, types of support and stress, to determine how these features impact not only individuals but also, potentially, their social network structure.

Recent developments in computer model integration have also demonstrated that ABMs which combine mechanisms from multiple theories can provide an improved explanation for complex phenomena (in terms of parsimony and empirical goodness-of-fit) [81,82]. These integration findings are particularly relevant given that theories of health inequality do not necessarily compete, but rather attempt to explain health inequality from different viewpoints.

Computer simulation methods such as ABM have yet to be applied to understand the AHP and would allow us to make best use of the available evidence to test the explanatory value of the mechanisms described in existing theories of health inequality. When we extract the mechanisms from these theories and implement them in an ABM simulation, does the simulation generate inequalities in alcohol-related harm?

## 4. Discussion

It is clear research investigating the AHP eschews the use of theory. Many of the mechanisms specified in health inequality theory are touched on as hypothetical explanations for the paradox, mainly on an ad-hoc basis and in the absence of clear theoretical structure. Structure would be provided by drawing on any of these theories explicitly. In the one instance where one of the theories was present in the empirical work on the AHP, this showed promise, as cumulative behaviours across the lifecourse could explain a greater proportion of harm experienced by lower socioeconomic groups [51]. There is a lack of evidence, which makes it difficult to conclude whether one theory over another can best explain the AHP, especially as these theories do not necessarily compete but examine causes of health inequality at different levels and with differing emphasis on certain factors. One thing is clear: the use of these theories will shift how we think about the causes of the paradox from health behaviour in isolation, to the wider context of complex interacting mechanisms between individuals and their environment.

Framing alcohol research using health inequality has significant implications for the study of the AHP and the wider alcohol harm research agenda. In the past, behavioural framings have resulted in empirical work underpinned by individual proximal factors, specifically alcohol consumption and other health behaviours. In Section 2, for each theory, we identified research designs implemented in social epidemiology which attempt to understand the causes of health inequality more generally (e.g., new measurements that capture social capital [50], or cross-national comparisons [68]). We can utilise the advances in social epidemiology, for example the introduction environmental resources in the materialist perspective [44] and apply this to the AHP.

Taking a behavioural approach has resulted in the implementation of policies which often rely solely on individuals taking action to reduce their alcohol consumption (e.g., educational campaigns), which arguably increase inequalities [15,83]. There have been attempts to reduce inequalities by introducing minimum unit pricing in several countries, including Scotland, Wales, and Australia’s Northern Territory. In theory this policy reduces the consumption of alcohol, particularly for those of a lower SEP, as they typically purchase alcohol at cheaper price points [84]. However, the focus of this policy remains on reducing alcohol consumption, which will not address the underlying causes of inequality.

Critically, shifting from this focus on alcohol consumption as the fundamental cause of harm in alcohol research requires researchers to acknowledge the causal processes driving harm are complex and that understanding of these processes requires different methodological perspectives, drawing on ideas from complexity science [85].

While the focus of this paper has been on the AHP, a well evidenced phenomenon, it is possible that a harm paradox could exist for other health behaviours. Hypothetically, with the same number of cigarettes smoked, those of a lower SEP may experience greater rates of smoking related harm, and there is evidence to support this hypothesis [86]. This reflects a slight misnomer—the AHP is not particularly paradoxical if it simply reflects wider causes of health inequalities. This concern further reinforces the need to utilise theories of health inequality to understand the complex interactions between health behaviour, the environment and harm and explore why lower socioeconomic groups are more vulnerable to the negative effects of risk behaviour.

## 5. Conclusions

The existing research on the causes of the AHP lacks theoretical structure and relies heavily on analysing the contribution of health behavioural risk factors. Drawing on health inequality frameworks would result in a more structured effort which gets at the root causes of both alcohol-related harm and alcohol-related health inequalities. Using these multi-level frameworks would allow us to understand the role of other mechanisms, in addition to alcohol consumption, which exist in the wider socioeconomic environment. Simulation methods (e.g., ABMs) allow for the opportunity to meaningfully explore the complexity captured in health inequality theory. Combining these theories with simulation methods has the potential to inform policy, which not only reduces consumption but also reduces harm, and in turn health inequalities, more broadly.

## Figures and Tables

**Table 1 ijerph-18-06025-t001:** Health inequality theories with descriptions.

Theory	Description
Social Determinants of Health	Contains four sub-theories (culture-behaviour, materialist, psychosocial and lifecourse). The social determinants of health specify the interacting role of factors from the narrowest sphere (e.g., individual biological mechanisms) to the broadest (e.g., the structure of society) [18]. These determinants can be distinguished into upstream factors (e.g., socioeconomic structure of society) and downstream factors, (e.g., individual factors, health policy and healthcare) [19,20]. The structures in society not only impact health directly but also indirectly by creating mechanisms (or SDH), which are then distributed to reflect the socioeconomic stratification of society [21].
Fundamental Cause Theory	Central to FCT are resources defined as money, knowledge, power, prestige, and social connections. It is proposed that high SEP groups have increased access to these flexible resources and can employ them to avoid risks, reduce the consequences of disease and uptake available treatment to improve health. Conversely, these resources are not readily available to low SEP groups. FCT opposes individualistic beliefs, emphasising that health cannot be individually controlled and is to some extent the responsibility of the state [22].
Political Economy of Health	The political economy account draws on the idea that cultural-behavioural, material, and psychosocial explanations are rooted in structures (e.g., politics, the economy, work, and labour markets) [23,24]. It is the wider macro-economic and political context that determines the distribution of the SDH, population health and inequalities [25,26]. This often occurs through public policy decision making, which is impacted by the corporate and business sector, labour, civil society, and political attitudes (e.g., individualistic versus environmentally or socially focused) [23].
Eco-social Model	The eco-social approach developed by Krieger is a multi-level theory which seeks to “develop analysis of current and changing population patterns of health, disease and well-being in relation to each level of biological, ecological and social organization” [27]. Key to this theory is the idea that biology and biological changes are determined by the social environment [10]. For example, alleged racial differences in biology (e.g., kidney function, blood pressure) posited by biomedical research are instead seen as the modifiable and embodied biological result of occupational and residential racial segregation [27].

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
