# Peer review of "Beyond Behaviour: How Health Inequality Theory Can Enhance Our Understanding of the ‘Alcohol-Harm Paradox’"

_ijerph, 2021, doi:10.3390/ijerph18116025_

Round 1

Reviewer 1 Report

This is a well-written study regarding the alcohol-harm paradox – the paper describes theories of health inequality to understand the paradox including the social determinants of health, fundamental cause theory, political economy of health and eco-social model. I find the research within this field relevant and important in relation to policy decisions.

I have few comments.

The paper could benefit from being shortened down. It is a long paper and it takes a lot of time to go through. I think it would benefit the readers if it was shorter and the most important things were highlighted and easier to find.  

Please provide a clear aim and a clear conclusion.

Please elaborate the part about computer simulations. As I understand you describe the benefit of using this technic, but you have not done them by yourself?

Reference no. 7 is a paper under review. My experience is that the International Journal of Environmental Research and Public Health is very fast at publishing papers, so if the paper is not yet accepted, it should be deleted from the references.

Author Response

Dear Reviewer 1,

We would first like to thank you for your thoughtful comments and efforts towards improving our manuscript. Please see a point-by-point response to the comments and concerns in the word document.

We hope our revisions and responses address the concerns raised.

Kind Regards,

Jennifer Boyd

Clare Bambra

Robin Purshouse

John Holmes

Reviewer 2 Report

ijerph-1214521

This is an ambitious and dense piece, which appears to be not just a review, but rather a commentary or think piece. It draws on epidemiological research to identify and review a number of theories that have been used to explain and study health inequalities. The paper has the potential to move the conversation and research forward on the alcohol harm paradox.

This review is different in that, because the paper is more conceptual and theoretical, I am giving feedback and “prompts” to help structure the argumentation and the presentation of the argumentation.

General and conceptual comments

As the paper is very dense, it could benefit from a clearer structure that is more concrete and more practical rather than being too detailed.

I believe that somewhere in the paper, the authors need to deal with the uniqueness of the AHP in that alcohol use can be both a risk factor and a harm. For example, heavy drinking or binge drinking can be seen as a harm on its own, and it is also a risk factor for interpersonal violence, accidents, etc. I believe that this must be kept in mind when trying to develop a model or a theory of AHP.

Related to this is also the need to consider whether a harm paradox can occur with another risk factor. A hypothetical example could be smoking: given the same number of smoked cigarettes would the poor have higher rates of lung cancer than the rich? How would case-finding (incidence, prevalence estimates) work in this instance? Or another example could be illegal drug use or mental health. My point is that if one is going to use theories drawn from social epidemiology, it might be worthwhile to consider how the theories work with other health behaviours in interaction with the environment, and whether a paradox could also exist elsewhere.

Finally, I found a number of places where subject-verb agreement was lacking, as well as some typos (e.g., singular vs. plural of some words) (also in the references, e.g. black Report). It would be beneficial if the authors had a third party proofread the text.

Abstract

There is some repetition in the abstract, especially after the authors introduce computer simulations. The authors could consider reformulating the final two sentences.

  1. Introduction

This section lays out in a clear way what the paper intends to do. However, in reading the subsequent sections, I became lost in the details and had difficulty to follow the “red thread”.

Page 2, line 44: I am not sure that the authors actually further develop the theoretical basis for AHP. Or, am I misunderstanding the aim of the paper? In listing and reviewing approaches and theories of social inequalities in social epidemiology, it is not stated clearly enough that the authors are actually proposing a concrete alternative approach to studying AHP.  Or are the authors proposing several ways by listing the various theories?  Perhaps it is also a matter of labelling the various sections in more specific ways.

By the way, it is a pity that reference [7] has not yet been published and could not be read in conjunction with the present manuscript.

  1. focusing on individuals…

This section could be part of the introduction. It does not necessarily need a separate heading.

  1. Theories of health inequality…

This is a very detailed section that lists and explains four theoretical approaches (and several sub-areas of the SDH approach). I believe that the text up to section 4 is in a way a sort of “throat clearing” and could afford to be shortened.

Nonetheless, the authors do mention one aspect of the psychological explanation of SDH line 137, page 3 “psychological experience of living in deprivation” (later referred to also as “social comparison”) which is a part of another area of research, namely, relative social inequalities or relative deprivation. This is an important concept, especially in high income social welfare state countries where social inequality persists. There area of research has its own literature.

  1. Theories of health inequality can broaden…

I regard this section as the main message of the paper. It also appears to be a critique of how the theories have been applied to alcohol research and AHP. Perhaps it should have a different title or heading? (Perhaps the paper should as well?) This section could be made more concrete with examples of how the theories could be applied in the future.

Thus, for each sub-section, 4.1 through 4.4, it seemed to me that the authors were offering generalized proposals as to how future alcohol research could apply these theories to help explain AHP. However, I was left feeling as if I was missing something. The authors had spent much time and effort to review and critique the theories, so that I was expecting that the authors themselves would propose a new synthesised approach and would propose the concrete testing of the new theory.

I realise that any good researcher would not want to give his or her newly developed theory away to others to test if such a researcher was planning to do that as well. I would be very interested to know where the authors stand with respect to this question. If it is the case that they are about to set out on some empirical work, then perhaps this paper should rather be integrated into such a future article?

Page 5, line 233: I am not sure what the sentence means: “while existing empirical work has focused on behaviour, there remains scope (??) to examine the interplay….” I am not sure what scope means in this context.

  1. computer simulations…

This section came across as a “teaser” to me, given what I have just mentioned above, as it seems that the authors are mentioning it because they intend to use such analyses in the future. If not, they could give more concrete examples of how it could be used, which I suggest anyway below.

Page 8, line 373: Please give a concrete example of how a computer simulation would work and what it would produce. Also more elaboration on how multilevel techniques would be included would also be helpful.

Page 8, line 389: Please expand on how the method would allow to test the explanatory value of mechanisms. For many readers, this will be their first introduction to this approach.

  1. Discussion

This section could be expanded somewhat to draw upon research that has taken place in social epidemiology in general and has tried to tackle the same sort of issue (see my general comments above). What are the implications from what has been learnt there?

  1. Conclusions

Page 9, line 418: What do “non-behavioural” framings” actually refer to? Do the authors mean non-individually based framings or analytical approaches; i.e., framings that consider the environment? The environment, whatever it is defined to consist of, is a basic element of public health research, along with the host (the individual) and the agent (here, alcohol). I think in this context “behavioural” framings are actually the same as individual framings, in which the individual is studied without consideration of his/her environment.

Author Response

Dear Reviewer 2,

We would like to thank you for your thoughtful comments towards improving our manuscript, which we have responded to comprehensively. Please see a point-by-point response to comments and concerns in the word document.

We hope our revisions and responses address the concerns raised.

Kind Regards,

Jennifer Boyd

Clare Bambra

Robin Purshouse

John Holmes

Reviewer 3 Report

This is a fantastic paper.

As I was reading it, I got to the end of section 3 and wrote a note that the theories were presented outside the context of alcohol-related harm. But when I read section 4, I realised that section is about applying these theories in alcohol-related harm. Had the authors presented that these two sections together, the reader would immediately see the application of these theories in context.

Author Response

Dear Reviewer 3,

We would first like to thank you for your thoughtful comments towards improving our manuscript.

In response to your comment regarding the structure of the paper we agree that it is important the reader can immediately see the application of these theories in context. We have now restructured the paper and what were previously sections 3 and 4 are now presented together as section 2.

We hope our revisions and responses address the concerns raised.

Kind Regards,

Jennifer Boyd

Clare Bambra

Robin Purshouse

John Holmes

Round 2

Reviewer 2 Report

I find the manuscript has improved substantially enough to warrant publication at this time.